# Metabolic Markers Associated with Progression of Type 2 Diabetes Induced by High-Fat Diet and Single Low Dose Streptozotocin in Rats

**DOI:** 10.3390/vetsci10070431

**Published:** 2023-07-02

**Authors:** Maria Andonova, Petko Dzhelebov, Krastina Trifonova, Penka Yonkova, Nikola Kostadinov, Krasimira Nancheva, Veselin Ivanov, Krasimira Gospodinova, Nikola Nizamov, Ilia Tsachev, Chavdar Chernev

**Affiliations:** 1Department of General and Clinical Pathology, Faculty of Veterinary Medicine, Trakia University, Stara Zagora 6000, Bulgaria; 2Department of Veterinary Anatomy, Histology and Embryology, Faculty of Veterinary Medicine, Trakia University, Stara Zagora 6000, Bulgaria; 3Clinical Laboratory, University Multiprofile Hospital for Active Treatment “Professor Stoyan Kirkovich”, Stara Zagora 6000, Bulgaria; 4Department of Social Medicine, Health Management and Disaster Medicine, Faculty of Medicine, Trakia University, Stara Zagora 6000, Bulgaria; 5Department of Veterinary Microbiology, Infectious and Parasitic Diseases, Faculty of Veterinary Medicine, Trakia University, Stara Zagora 6000, Bulgaria; 6Independent Researcher, Surrey CR3 5ZD, UK

**Keywords:** rats, diabetes type 2, high-fat diet, streptozotocin, insulin resistance, obesity, metabolic markers

## Abstract

**Simple Summary:**

Our experimental model of T2DM in rats is based on the development of insulin resistance induced by a four-week high-fat diet leading to obesity and pancreatic β-cell dysfunction caused by streptozotocin (STZ). Changes in anthropometric parameters, glucose, insulin, lipids, uric acid, advanced oxidation protein products (AOPP), as well as the histological changes in the liver and pancreas, were recorded. To assess insulin resistance, we used the Homeostasis Model Assessment of Insulin Resistance (HOMA-IR) and beta cell function (HOMA-β) and visceral obesity—adiposity index (AI). The data demonstrate that the increasing values of glucose, HOMA-IR, AI, total cholesterol, triacylglycerols, low- and very-low-density lipoproteins are important markers of the pre-diabetic state induced by a long-term intake of a high-fat diet. Early metabolic markers of the disease were persistent hyperglycemia and hypercholesterolemia, as well as increased uric acid, HOMA-IR and decreased HOMA-β. The progression of T2DM in rats was characterized by hyperglycemia, hypercholesterolemia, hypertriacylglyceridaemia, an increase in lipoproteins, AI, uric acid, AOPP, HOMA-IR and decreased HOMA-β. A precise combination of several easily accessible metabolic markers is necessary to provide accurate information about T2DM in rats.

**Abstract:**

Science is still searching for readily available, cost-effective biomarkers to assess metabolic disorders occurring before the onset and during the development of type-2 diabetes (T2DM). The aim of the present study was to induce T2DM in rats through a high-fat diet, followed by a single administration of low dose streptozotocin (STZ), and make an assessment of the development of the disease. The rats were divided into two groups—experimental and control—and were monitored for a period of 10 days. Changes in anthropometric parameters, glucose, insulin, lipids, uric acid, advanced oxidation protein products (AOPP), as well as the histological changes in the liver and pancreas, were recorded. To assess insulin resistance, we used the Homeostasis Model Assessment of Insulin Resistance (HOMA-IR) and beta cell function (HOMA-β) and visceral obesity—adiposity index (AI). The data demonstrate that the increasing values of glucose, HOMA-IR, AI, total cholesterol, triacylglycerols, low- and very-low-density lipoproteins are important markers of the pre-diabetic state. The stable hyperglycemia and increased levels of TC, TG, VLDL, LDL, uric acid and AOPP in experimental rats strongly suggest the development of T2DM. HOMA-IR, HOMA-β, AI, and uric acid are reliable criteria for T2DM in rats.

## 1. Introduction

Type 2 diabetes (T2DM) is a polyetiological, chronic, metabolic disease in different biological species [1,2,3]. Usually, in the early stages, this disease is asymptomatic, but the resulting disorders of carbohydrate, lipid and protein metabolism determine the later clinical signs (polyuria, polydipsia and polyphagia). Pancreatic β-cell dysfunction and insulin resistance play an essential role in the development of T2DM [4]. Insulin resistance can be provoked by high levels of non-esterified fatty acids, glycerol, hormonal imbalance, hypoxia, oxidative stress and genetic factors. However, the main predisposing factor for insulin resistance is obesity, which is involved in the pathogenesis of T2DM [5]. Typically, the diagnosis of T2DM is largely based on anthropometric characteristics and blood panels measuring insulin, glucose, glycated hemoglobin and lipid metabolites. Recently, however, it has been commented that some of the conventionally used markers for diabetes—body mass index (BMI) and glycated hemoglobin (HbA1c)—do not provide accurate information because they do not take into account the influence of genetic, species, population and anthropometric parameters [5,6]. As an alternative to BMI, many experts recommend measuring the visceral adipose tissue [7]. Evaluation of glycemic variability in well-controlled type 2 diabetes mellitus may require confirmation with other biomarkers [8,9]. Different genomics, metabolomics, proteomics and microbiomics and RNA sequencing-based studies are new revolutionizing biomarkers for diabetes risk, but their determination is costly and time consuming [10,11,12,13]. Scientists are still searching for biomarkers and methods for screening and early diagnosis of T2DM, the determination of which is easy, fast and does not require complex equipment, as well as significant financial costs [12,13,14]. There are increasing attempts to improve the available algorithms for T2DM by including indices such as quantitative insulin sensitivity check index (QUICKI), homeostatic model assessment (HOMA), adiposity index (AI) [15,16] and metabolic biomarkers to identify pre-diabetes and T2DM [17,18]. The integration of biomarkers reflecting different pathogenetic mechanisms of the disease allows to differentiate key metabolic pathways related to purine nucleotides and their metabolites and oxidative stress [19,20,21].

Rat models are widely used to study the metabolic profile of some aspects of the pathology of T2DM. Rats have a short gestation period, reach sexual maturity quickly [22,23,24], easily adapt to different diets, environmental conditions and manipulations [25]. A good experimental model elucidating the role of obesity and insulin resistance as leading pathogenetic mechanisms in T2DM is the one involving a combination of a high-fat diet [26] with the administration of a subdiabetogenic dose of streptozotocin (STZ). STZ is a highly selective pancreatic islet β-cell-cytotoxic agent [27]. However, there are various modifications to this experimental model, affecting both the duration and composition of the high-fat diet (HFD), as well as the STZ dose and route of administration [28]. Despite the existence of a large number of studies related to the modeling of T2DM in rats by means of HFD and STZ, there are no sufficiently clear criteria for evaluating the different stages of the progression of diabetes in this biological species. Skovsø [29] extensively discussed the specific variations in dietary regimen, age and dose of STZ when using HFD/STZ rat models. This researcher also summarized metabolic parameters (insulin, glucose, total cholesterol, triglycerides, LDL, HDL, HOMA-IR, HbA1c) used to identify the stage of T2DM in rats. Researchers must choose appropriate models, according to the different requirements of their studies, to obtain credible research results because the pathogenesis of diabetes is complex [30,31]. Although Perlman [32], Bahadoran et al. [33], Wen et al. [34] have discussed the limitations of experimental models, studies on animals cannot be fully replaced by in vitro methods. The exploration of physiological functions and systemic interactions between organs, as is the case for most hormonal regulations, makes investigations in animals still necessary to support the hypotheses suggested by studies on separate tissues and organs. In rats, however, there are no uniform criteria for evaluating T2DM, and the emphasis on BMI as an indicator of obesity in this biological species does not always correspond to the degree of obesity induced by the applied diet [35]. Accurate information about the dynamics of disorders occurring before and after the onset of T2DM in Wistar rats, as well as readily available biomarkers for its assessment, are necessary.

The aim of the present study was to induce T2DM in male Wistar rats through a 4-week high-fat diet, followed by a single administration of low dose STZ, and to use this experimental model to make a integral assessment of the development of the disease by tracking the changes of anthropometric parameters (body weight, body length, BMI, abdominal circumference), some indices (HOMA-IR, HOMA-β, AI), metabolic parameters of carbohydrate and lipid metabolism, purine metabolites, protein products of advanced oxidation (AOPP) and histologic changes of the liver and pancreas in order to select the most reliable evaluation criteria suitable for this biological species.

## 2. Materials and Methods

### 2.1. Animals

Male Wistar rats (*n* = 70), 6–7 weeks of age, with an initial body weight of 160–180 g were used in the study. Animals were housed at 22 °C ± 2 °C, controlled humidity 55 ± 10%, 12:12 h light-dark photoperiod and had access to water and food ad libitum. After an adaptation period of 2 weeks, the rats were divided into 2 groups: control group (*n* = 35) and experimental group (*n* = 35).

All animal experiments were in accordance with the ethical standards (Permit No. 281 of the Bulgarian Food Safety Agency; opinion of the Ethics Committee No. 197 of 10. 09. 2020).

### 2.2. Diets

Animals from the control group received ad libitum standard food for laboratory rats (HL-TopMix) throughout the experimental period. 100 g of food contained 332 kcal (6% of calories are from fat). The component composition of the food for laboratory animals (TopMix) included wheat, corn, sunflower meal, wheat bran, calcium carbonate, bonding agent, sodium chloride, premix and nutritional supplements.

During the adaptation period, rats from the experimental group were also fed a standard diet. During the experimental period, they received a high-fat diet for a period of 4 weeks. Fat in the high-fat diet provided 43% of the energy intake. The high-fat food was prepared in situ with the addition of lard to the standard food. Thus, 100 g of the resulting food (high-fat diet) contained 445.6 kcal.

### 2.3. Diabetes Induction

After 4 weeks on a high-fat diet, rats from the experimental group were left to fast for one night, after which they received a single subdiabetogenic dose of STZ (Sigma Aldrich, St. Louis, MO, USA) intraperitoneally (i. p.)—35 mg/kg body weight freshly dissolved in freshly prepared citrate buffer pH = 4.5 [36]. All STZ-injected rats (*n* = 35) had blood glucose ≥ 12 mmol/L and were considered diabetic [37] and were used in the present study.

### 2.4. Dynamics of Sampling

Rats were fasted overnight before blood sampling by retro-orbital bleeding. In experimental rats, blood samples were collected at the initial level, after 4 weeks on a high-fat diet and on days 1, 3, 5 and 10 after STZ administration. At each time point, single blood samples were obtained from 7 rats. In control rats, blood samples were collected at the same time points. The following parameters of carbohydrate, lipid, protein and purine metabolism were determined:Insulin—by Rat Insulin ELISA Kit (Cat No: RTFI00920, AssayGenie, Dublin, Ireland);Blood glucose—by enzymatic colorimetric test for glucose method without deproteinisation (Human GmbH, Wiesbaden, Germany);Total cholesterol (TC), triglycerides (TG), LDL, HDL—biochemical analyzer IDEXX Vet Test, USA;VLDL—level was calculated from Friedewald’s formula: VLDL = TG/5 [38];Uric acid—biochemical analyzer IDEXX Vet Test, USA;Advanced oxidation protein products (AOPP)—spectrophotometric method [39].

All assays were performed in a certified lab in compliance with good laboratory practices and internal quality control procedures.

Homeostatic model assessment:HOMA-IR was determined on the basis of the obtained results from the measurement of insulin and fasting blood sugar using the formula HOMA-IR = glucose × insulin/22.5 [40];HOMA-β was determined by the formula 20 × insulin/glucose − 3.5.

The anthropometric parameters were measured for each group in the same dynamics, but without days 1 and 3.

Body weight (BW)—the body weight of each animal was measured using an electronic scale (OHAUS™ Scout™ STX, Ohaus Corporation, Northglenn, CO, USA);Body length—the length of the animals was determined by measuring the distance from the nose to the anus;Body mass index (BMI)—was determined using the formula body weight (g)/body length (cm)^2^ [41];Abdominal circumference—the circumference of the abdomen at its widest part was measured using a measuring tape;Adiposity index—at each stage of the study, 5 animals from each group were anesthetized with sodium pentobarbital (50 mg/kg BW, i. p.) and decapitated. Inguinal, epididymal and perirenal fat depots were precisely dissected and their absolute mass (in grams) was determined using an electronic scale (OHAUS™ Scout™ STX, Ohaus Corporation, Northglenn, CO, USA). The adiposity index was calculated as total body fat/BW × 100 [42].

At each stage of the study, material for histological examination was taken from the liver and pancreas of rats from the studied groups. The collected samples were fixed in a 10% neutral formaldehyde solution for 48 h. After washing and dehydrating the tissue samples in the ascending alcohol series, they were cleared and embedded in paraffin. Serial histological sections of 3 to 5 µm thickness were obtained with a Leica RM 2235 rotary microtome (Leica Microsystems, Nussloch, Germany) and stained with hematoxylin/eosin. Microscopic observations were performed with a Leica DM1000 -LED light microscope equipped with the Leica Application Suite software platform (LAS, version 4.8.0., Leica Microsystems CMS GmBH, Heerbrugg, Switzerland).

### 2.5. Statistical Analysis

Data are expressed as mean ± standard deviation (SD). Statistical analysis was performed using one way analysis of variance (One-way ANOVA) followed by Tukey’s post Hoc test using Graph Pad InStat 3.1 Software. For all analyses, the level of statistical significance was set at *p* < 0.05.

## 3. Results

### 3.1. Anthropometric Parameters

At the start of the experiment, the body weight of the rats included in the control and experimental groups was similar—160.57 ± 18.96 g and 175.14 ± 20.39 g, respectively. After 4 weeks on a standard diet, the body weight of rats from the control group increased to 271.43 ± 26.70 g, which was statistically significant compared to the initial weight (*p* < 0.001). In this group, the trend towards weight gain continued until the end of the study on the 10th day, when it reached 299 ± 30.54 g (*p* < 0.001 compared to baseline). The body weight of rats from the experimental group after 4 weeks on a high-fat diet reached 295.14 ± 31.02 g. This value was close to the weight reported in controls in the same period, with no statistically significant difference between the two groups. In the experimental group on day 5 after STZ administration, the weight was slightly reduced (287.86 ± 27.93 g); on the 10th day, it was 292.43 ± 29.47 g, and the deviations were not statistically significant compared to the control group. Rats from both groups started with a body mass index ranging between 0.44 ± 0.05–0.49 ± 0.06 g/cm^2^. This indicator, regardless of the diet— standard or high-fat—showed a statistically significant increase compared to the initial level (*p* < 0.001). At the end of the experimental period on the 10th day, BMI in experimental rats decreased to 0.57 ± 0.04 g/cm^2^, and the difference was statistically significant compared to controls (*p* < 0.05). The data in Table 1 shows that in rats from both groups, naso-anal distance and abdominal circumference were the anthropometric parameters that, regardless of the nature of the diet, increased gradually, and their values were similar and there was no statistically significant difference between the groups.

### 3.2. Parameters for Determination of Insulin Resistance in Rats

The data presented in Table 2 for the serum concentrations of insulin and glucose show that the values of these indicators in the animals of the control group varied around the baseline levels, both after the 4-week standard diet and until the end of the study. In contrast, rats from the experimental group, after 4 weeks of the high-fat diet, responded with a statistically significant increase in insulin (*p* < 0.01 vs. baseline) and glucose (*p* < 0.05 vs. control). In these animals, after STZ injection, glucose concentrations continued to rise until the end of the study. On the 1st day after STZ, blood sugar showed levels significantly higher compared to the initial level (*p* < 0.001) and after diet (*p* < 0.001), as well as compared to controls. The blood sugar increased progressively and, at the end of the study, reached its highest value (27.55 ± 5.59 mmol/L). In the experimental group, insulin peaked on day 3 (17.05 ± 2.73 μIU/mL, *p* < 0.01 versus initial level). After this period, its values decreased, being 14.26 ± 12.59 μIU/mL on the 5th day and 14.60 ± 3.80 μIU/mL on the 10th day, but these deviations did not have statistical significance.

In HOMA-IR, the most significant changes were registered in the rats of the experimental group. The 4-week high-fat diet led to an increase that was statistically significant compared to animals receiving the standard diet (*p* < 0.05) (Figure 1). After STZ treatment, HOMA-IR increased, and the trend was sustained on days 1, 3, 5, 10 (*p* < 0.01 vs. controls).

In these animals, insulin resistance was accompanied by β-cell dysfunction, as evidenced by the low HOMA-β values recorded from day 1 after STZ treatment until the end of the study on day 10 (Figure 2). The deviations of this parameter were statistically significant and markedly lower, not only compared to the initial level (*p* < 0.001), but also compared to after diet (*p* < 0.001) and the control group (*p* < 0.001).

The adiposity index in the rats of the experimental group increased after the high-fat diet and reached a maximum on the 5th day after STZ, followed by a decrease on the 10th day (Figure 3).

### 3.3. Serum Lipid Profile

The lipid profile data of rats from the study groups presented in Table 3 indicate that diet is the factor that significantly influences serum concentrations of TC and TG and lipid metabolites (VLDL, LDL, HDL). In controls rats, values of the lipid profile parameters varied around the baseline through the entire study period. In contrast to that, lipid profile parameters of experimental rats demonstrated a statistically significant increase after four weeks on a high-fat diet compared to the controls: TC (*p* < 0.001), TG (*p* < 0.001), VLDL (*p* < 0.001), LDL (*p* < 0.001), as well as HDL (*p* < 0.05) (Table 3). After this period, the TC concentrations of the experimental group increased, and at the end of the study—the 10th day—they reached their highest value (3.07 ± 1.85 mmol/L).

In experimental rats, an increase in TG, VLDL, LDL was also recorded, and the trend was well expressed until day 5, when TG reached 8.32 ± 4.57 mmol/L (*p* < 0.001 vs. controls; *p* < 0.001 vs. initial level; *p* < 0.01 vs. after diet), VLDL reached 3.81 ± 2.09 mmol/L (*p* < 0.001 vs. controls; *p* < 0.001 vs. initial level; *p* < 0.01 vs. after diet), and LDL reached 0.89 ± 0.45 mmol/L (*p* < 0.001 vs. controls; *p* < 0.001 vs. initial level). On the 10th day, the values of these parameters decreased slightly: TG to 5.95 ± 3.99 mmol/L (*p* < 0.05 compared to controls; *p* < 0.05 compared to initial level), VLDL to 2.72 ± 1.82 mmol/L (*p* < 0.05 vs. controls; *p* < 0.05 vs. baseline), LDL to 0.76 ± 0.67 mmol/L (*p* < 0.05 vs. day 3). The deviations in serum HDL concentrations in the experimental animals after the high-fat diet were not statistically significant (Table 3).

### 3.4. Blood Protein and Purine Markers

The data presented in Table 4 shows that the serum concentrations of advanced oxidation protein products (AOPP) in rats from the experimental group increased, statistically significantly, on the 3rd day, when the deviations were significant, not only compared to initial level (*p* < 0.001), but also compared to controls (*p* < 0.001). On day 5, values reached a maximum of 181.00 ± 44.38 μmol/L (*p* < 0.001 vs. initial level, *p* < 0.001 vs. diet, *p* < 0.001 vs. controls), remaining high until the end of the study—the 10th day. Compared to the control group, uric acid values in diabetic rats increased, statistically significantly, as early as day 1 after STZ, reached a peak on day 3 (*p* < 0.05 vs. controls) and remained high till the end of the study (Table 4).

### 3.5. Histological Changes

In comparison to the normal histostructure of liver in rats from the control group (Figure 4), the results of the light microscopic observations showed that after the end of the high-fat diet, in some areas, the liver parenchyma had a changed structure, expressed in an increase in hepatocyte sizes, a change in their shape and the accumulation of fat droplets in the cytoplasm. Fat vacuoles were mainly located perinuclearly. The nuclear membrane and nucleoli were intensely stained and clearly visible against the background of vacuoles in the cytoplasm (Figure 5a). On the 5th day after STZ application, the liver parenchyma had a preserved configuration of the lobules with normal radial orientation of the plates. It was established congestion and dilatation of central veins, as well as dilated sinusoidal capillaries in the three zones of the liver lobules.

In periportal zones, hepatocytes had undergone fatty degeneration and the nuclei were pyknotic. In perivascular areas, we observed leukocyte infiltrations (predominantly neutrophils) and an increasing number of Kupffer cells (Figure 5b).

Histological investigations on the 10th day after STZ application showed a normal structure of the liver lobules with a low degree of leukocyte infiltration of the periportal areas and faintly visible dilatations of the sinusoidal capillaries (Figure 5c).

Healthy rats showed normal histological pancreas characteristics (Figure 6).

It was found that after the end of the high-fat diet, the pancreas had a preserved microstructure. In close proximity to the interlobular excretory ducts and blood vessels, we observed compact groups of white adipocytes (Figure 7a). On the 5th day of administration of STZ, ubiquitously dilated intralobular and interlobular ducts were found in the pancreatic parenchyma. There were sections with well-preserved endocrine and exocrine parts of the parenchyma, but in other areas, the islets of Langerhans were distorted in shape, contour and density, or absent altogether (Figure 7b). On the microscopic slides of the pancreas of 10-day STZ rats, slightly dilated intralobular and interlobular excretory ducts were established. Most of the parenchyma showed well-preserved histostructure and the presence of small islets of Langerhans. At higher magnifications, the leukocyte infiltration (predominantly neutrophils) around them was visible. In some lobules, there was lack islets of Langerhans (Figure 7c).

## 4. Discussion

Our experimental model is based on the development of insulin resistance induced by a four-week high-fat diet leading to obesity and pancreatic β-cell dysfunction caused by a single subdiabetogenic dose of STZ. This experimental model emphasizes the role of the leading pathogenetic mechanisms of this disease—insulin resistance in obesity [30] combined with impairment of pancreatic β-cell functions [31]. Unlike obesity in humans, which requires a long period of time, in rats, obesity can be induced in a few weeks by hypercaloric diets rich in carbohydrates (high-carbohydrate diet) or fats (high-fat diet) or a combination of the two nutritional components (high-fat, high-carbohydrate diet) [43]. In our experimental model, a high-fat diet was used—lard was added to the diet of the experimental animals so that fat provided 43% of the energy intake. According to Buettner et al. [44], when a high-fat diet is based on the intake of lard containing both saturated and monounsaturated fatty acids, the results achieved are better than diets rich in coconut or fish oil. Of course, the choice of diet, its composition and duration are important for inducing obesity in rats, but there are some considerable difficulties due to the lack of clear criteria for the characterization of rat obesity [35]. The advantage of our model is the short duration of the high-fat diet. Some studies use an even shorter period of a high-fat diet, but without changes in the blood glucose concentrations [36]. Our study demonstrates that 4 weeks on a high-fat diet (calories from fat > 40%) can disrupt glucose metabolism. Moreover, high-fat diets with fat higher than 50% have an influence on the severity of metabolic disorders and β-cellular function [45]. Obesity is an important predisposing factor for the development of type 2 diabetes. Pre-diabetes often includes a state of obesity characterized by insulin resistance and dyslipidemia. Essentially, however, the transition from a pre-diabetic state to overt diabetes requires the loss of a significant proportion of functional β-cell mass [29].

The anthropometric parameters presented in Table 1 (body weight, length, abdominal circumference) are easily accessible and measured, but the analysis of their values shows that their increase was statistically significant compared to initial level, both in controls and in the experimental animals, and there were no statistically significant deviations registered between the two groups during the entire study period. The calculated BMI (Table 1) has baseline values ranging between 0.44 ± 0.05–0.49 ± 0.06 g/cm^2^, which correlates with the data of Novelli et al. [41] for this parameter in male Wistar rats, but changes were not indicative of obesity in the rats fed a high-fat diet. Most likely, changes in anthropometric indicators are the natural consequence of animal growth. According to Ghasemi et al. [16] and Hoffman et al. [46], rats have a high growth rate by the 14th week (98 days) after birth, although around day 60 they are already sexually mature [47]. The similarity we registered in the values of the anthropometric parameters in the control and experimental animals shows that they are not accurate markers of obesity in this biological species. According to us, the most objective criterion for the actual degree of obesity in the rats of the experimental group is the calculation of AI (Figure 3), which is based on the calculation of both body weight and the weight of inguinal fat + epididymal fat + perirenal fat after rat dissection. The baseline values of 4.08 ± 0.18% of this parameter in male Wistar rats correspond to those obtained by Leopoldo et al. [15]. Under the conditions of our experiment, the values increased both after the high-fat diet and after STZ injection (Figure 3). Therefore, the calculation of AI and the evaluation of visceral fat are important and reliable indicators of obesity in this biological species, especially in contrast to the recently used BMI. Kanasaki and Koya [48] also consider AI as an indicator of fat accumulation. Dual energy X-ray absorptiometry (DEXA) is currently widely used in humans as one of the most precise noninvasive methods for analysis of body composition. Cole et al. [49] has demonstrated the use of DEXA in mice.

Wondmkun [50] emphasizes that obesity is a triggering factor for diabetes associated with insulin resistance, in which the biological effects of insulin in physiological concentrations are impaired [30,31]. The dynamics of changes in the serum concentrations of this hormone in our experimental model of T2DM demonstrate inadequate insulin regulation. Regardless of its increased concentrations (after the high-fat diet, on the 1st and 3rd day after STZ), glucose values increased dramatically until the end of the study (Table 2). The increasing values of HOMA-IR (Figure 1) prove the presence of insulin resistance in the rats from the experimental group. After the high-fat diet HOMA-IR reached 5.06 ± 1.44, the trend continued until the 10th day after STZ administration, when this parameter reached a value of 14.80 ± 8.35 (*p* < 0.01 compared to controls). In their research, Antunes et al. [51] and Chao et al. [52] found that HOMA-IR directly correlates with the insulin tolerance test and is an accurate marker for detecting insulin resistance in Wistar rats. Against the background of progressively increasing values of HOMA-IR in the experimental animals, the reduction of HOMA-β is clearly evident. Based on the fact that HOMA-β characterizes the secretory capacity of β-cells of the pancreas, it becomes clear that β-cell function is reduced after administration of STZ (Figure 2). However, decreased pancreatic β-cell secretion is relative and not absolute, which is confirmed on one hand by the insulin values, which on the 10th day after the induction of diabetes are comparable to those of the controls (Table 2), and on the other hand by the histological findings, which show areas in the pancreas with well-preserved endocrine and exocrine parts of its parenchyma and others with destruction of the islets of Langerhans (Figure 7b,c). STZ, which in our experimental model was administered in a low dose (35 mg/kg body weight), is an agent that selectively destroys the pancreatic β-cells [28], which decreases the synthesis and secretion of insulin. STZ is one of the most commonly used diabetogenic agents because it is cheap and has fewer side effects. Its selective toxic effect on β-cells of the pancreas is related to both alkylation of DNA, resulting in fragmentation of the DNA, and generation of reactive oxygen species [53]. Insulin resistance provokes dysregulation, not only of glucose, but also of lipid metabolism [30,54]. In rats fed a high-fat diet, we observed opalescence of the sera on days 5 and 10 after STZ administration, which may be related to an increase in chylomicron concentrations in the blood and to insufficient activity of lipoprotein lipase [31]—a key enzyme for chylomicron degradation and removal from the circulation, as well as for lipolysis of the VLDL synthesized by the liver. The well pronounced hypertriacylglyceridaemia on the 5th and 10th day in experimental rats is accompanied by hypercholesterolemia and an increase in VLDL and LDL (Table 3). Changes in the lipid profile, according to Schenk et al. [55] and Yeasmin et al. [56], are associated with an impaired ability of insulin to inhibit hormone-sensitive lipase and an increase in serum concentrations of free fatty acids. High levels of free fatty acids induce insulin resistance in adipose tissue and skeletal muscle [31]. At the same time, the liver absorbs the free fatty acids, which are stored in the form of triacylglycerols in the fat droplets. This is confirmed by the histological changes in the liver of rats from the experimental group—in the periportal areas we found fatty degeneration of hepatocytes (Figure 5b,c). The in-depth analysis of the lipid profile in these animals (Table 3) also demonstrates that the intake of high-fat food is a factor disrupting the lipid metabolism, which is confirmed by the statistically significant increase in TC, TG, VLDL, LDL. This can be considered as an indication of a pre-diabetic state. This statement is supported by the increase in HOMA-IR recorded in the experimental rats (Figure 1). Therefore, dietary-induced insulin resistance in rats develops rapidly—after 4 weeks of a high-fat diet. Uric acid, which is an end product of purine metabolism in the liver, is included in the assessment of developing diabetes in rats. Ottosson et al. [57], Concepcion et al. [58] and Anothaisintawee et al. [59] commented that increased uric acid concentrations provoke oxidative stress, which is responsible for pancreatic β-cell dysfunction and endothelial dysfunction involved in the development of diabetes. Our data show that serum uric acid concentrations in diabetic rats increased on the 1st day of STZ administration and remained at a high level until the end of the study (Table 4). Therefore, uric acid is an early indicator of diabetic progression in the animals of the experimental group. The increased level of uric acid is related to the abnormal glucose metabolism and provokes oxidative stress, which is an important player in the pathogenesis of T2DM. AOPP are another important indicator of oxidative stress. Our data clearly show that AOPP increase dramatically on day 3 of the development of diabetes, when they are significantly higher, not only compared to controls, but also compared to their initial levels (*p* < 0.001). Values reached a maximum on day 5 (*p* < 0.001) and remained high on day 10 (*p* < 0.001) (Table 4). The increased AOPP can be informative to protein damage. Conti et al. [20] have also recorded increased concentrations of AOPP in diabetic patients.

## 5. Conclusions

Our experimental model is based on the development of insulin resistance induced by a four-week high-fat diet leading to obesity and pancreatic β-cell dysfunction caused by a single subdiabetogenic dose of STZ. This model allows analyzing the dynamics of the developing biochemical and histological changes, which characterize the progression of the disease. The data demonstrate that the increasing values of glucose, HOMA-IR, AI, total cholesterol, triacylglycerols, low- and very-low-density lipoproteins are important markers of the pre-diabetic state induced by a long-term intake of a high-fat diet. Early metabolic markers of the disease were persistent hyperglycemia and hypercholesterolemia, as well as increased uric acid, HOMA-IR and decreased HOMA-β. Progression of T2DM in rats was characterized by hyperglycemia, hypercholesterolemia, hypertriacylglyceridaemia, an increase in lipoproteins, AI, uric acid, AOPP, HOMA-IR and decreased HOMA-β. A precise combination of several easily accessible metabolic markers is necessary to provide accurate information about T2DM in rats.

## Figures and Tables

**Figure 1 vetsci-10-00431-f001:**
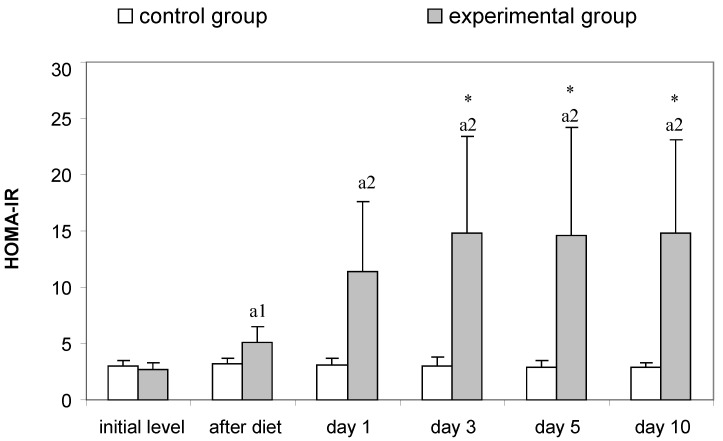
HOMA-IR in control and experimental group (*n* = 7). Results are presented as mean ± SD. Statistically significant differences within groups (compared to initial level)—* *p* < 0.05; statistically significant differences between groups (compared at the same points of dynamics)—^a1^ *p* < 0.05; ^a2^ *p* < 0.01.

**Figure 2 vetsci-10-00431-f002:**
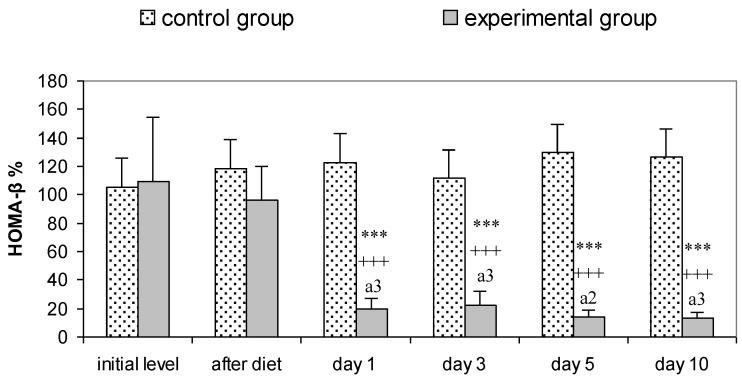
HOMA-β in control and experimental group (*n* = 7). Results are presented as mean ± SD. Statistically significant differences within groups (compared to initial level)—*** *p* < 0.001; compared to after diet period—^+++^ *p* < 0.001; statistically significant differences between groups (compared at the same points of dynamics)—^a2^ *p* < 0.01; ^a3^ *p* < 0.001.

**Figure 3 vetsci-10-00431-f003:**
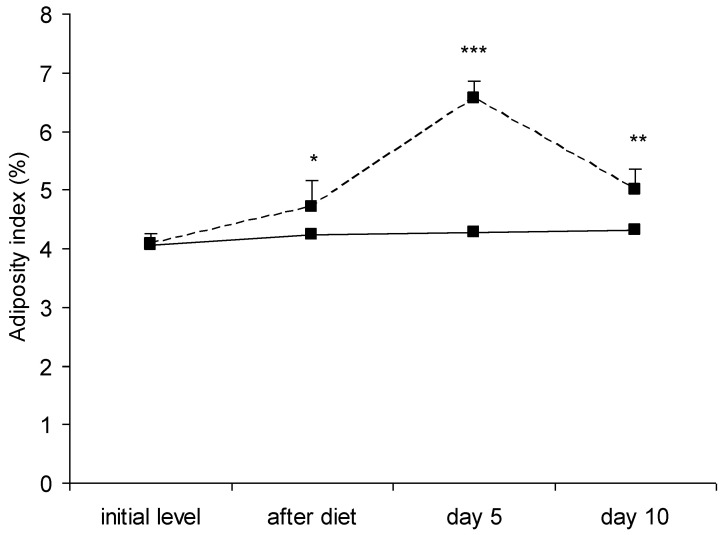
Dynamics of changes in the adiposity index in rats of control group (continuous line) and experimental group (dotted line). Statistically significant differences within groups (compared to initial level)—* *p* < 0.05; ** *p* < 0.01; *** *p* < 0.001.

**Figure 4 vetsci-10-00431-f004:**
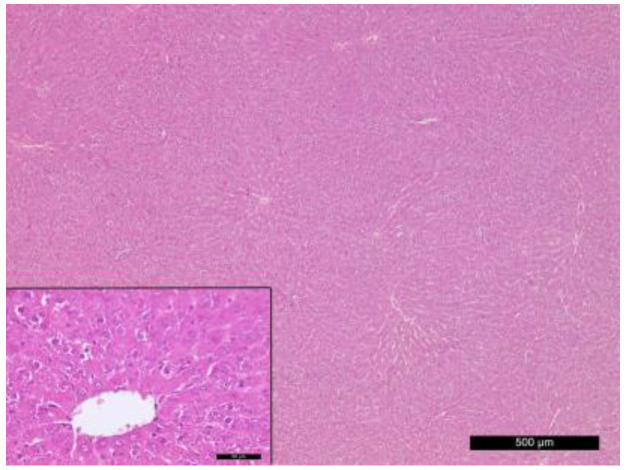
Normal histostructure of liver in rats from control group. Magnification 50× and 400× in the small box.

**Figure 5 vetsci-10-00431-f005:**
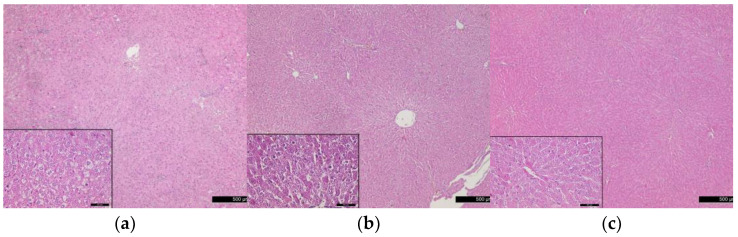
Changes in liver histostructure in rats from experimental groups (**a**) after the end of the high-fat diet, (**b**) on the 5th day after the STZ application, (**c**) on the 10th day after the STZ application. Magnification 50× and 400× in the small boxes.

**Figure 6 vetsci-10-00431-f006:**
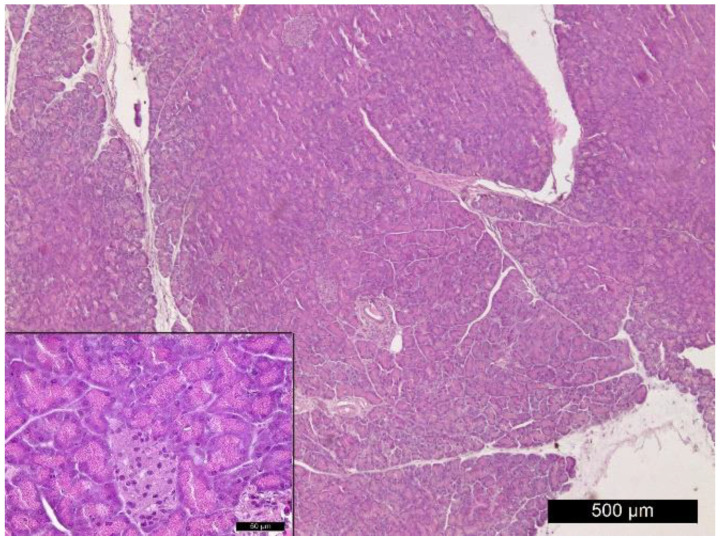
Normal histostructure of pancreas in rats from control group. Magnification 50× and 400× in the small box.

**Figure 7 vetsci-10-00431-f007:**
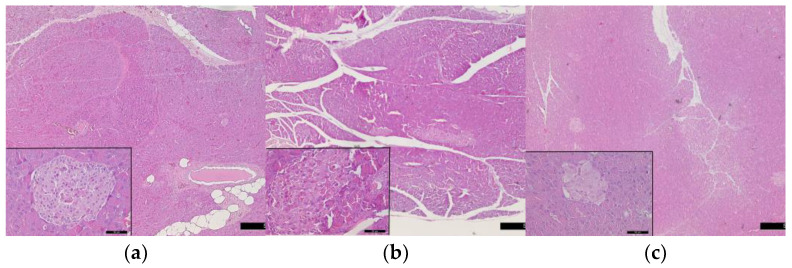
Changes in pancreas histostructure in rats from experimental groups (**a**) after the end of the high-fat diet, (**b**) on the 5th day after the STZ application, (**c**) on the 10th day after the STZ application. Magnification 50× and 400× in the small boxes.

**Table 1 vetsci-10-00431-t001:** Anthropometric parameters (body weight, body length, body mass index, abdominal circumference) in control group (*n* = 7) and experimental group (*n* = 7). Results are presented as mean ± SD.

Parameter	Group	Initial Level	After Diet	Day 5	Day 10
Body weight (g)	Control	160.57 ± 18.96	271.43 ± 26.70 ***	281.00 ± 23.45 ***	299.57 ± 30.54 ***
	Experimental	175.14 ± 20.39	295.14 ± 31.02 ***	287.86 ± 27.93 ***	292.43 ± 29.47 ***
Body length (cm)	Control	19.00 ± 0.58	21.00 ± 0.82 ***	21.71 ± 0.76 ***	21.71 ± 0.95 ***
	Experimental	19.35 ± 0.60	21.86 ± 1.07 ***	21.14 ± 1.68 **	22.71 ± 0.49 ***^a1^
BMI (g/cm^2^)	Control	0.44 ± 0.05	0.62 ± 0.05 ***	0.60 ± 0.04 ***	0.64 ± 0.06 ***
	Experimental	0.49 ± 0.06	0.62 ± 0.05 **	0.65 ± 0.09 ***	0.57 ± 0.04 ^a1^
Abdominal circumference (cm)	Control	15.28 ± 0.76	16.86 ± 0.69 ***	17.00 ± 0.58 ***	17.42 ± 0.58 ***
	Experimental	15.32 ± 0.60	16.42 ± 0.78 *	17.57 ± 1.27 *	17.72 ± 1.11 *

Statistically significant differences within groups (compared to initial level)—* *p* < 0.05; ** *p* < 0.01; *** *p* < 0.001; statistically significant differences between groups (compared at the same points of dynamics)—^a1^ *p* < 0.05.

**Table 2 vetsci-10-00431-t002:** Insulin and glucose in control and experimental group (*n* = 7). Results are presented as mean ± SD.

Parameter	Group	Initial Level	After Diet	Day 1	Day 3	Day 5	Day 10
Insulin (μIU/mL)	Control	11.33 ± 1.28	12.76 ± 1.49	12.02 ± 0.90	11.64 ± 2.30	11.98 ± 1.74	11.16 ± 1.04
	Experimental	10.80 ± 0.86	16.85 ± 2.70 **	15.44 ± 3.24 *	17.05 ± 2.73 **^a1^	14.26 ± 12.59	14.60 ± 3.80
Glucose (mmol/L)	Control	5.81 ± 0.66	5.74 ± 0.52	5.81 ± 1.07	5.7 ± 0.55	5.80 ± 1.01	5.84 ± 1.20
	Experimental	5.80 ± 1.01	7.20 ± 1.13 ^a1^	19.97 ± 4.29 ***^+++a3^	21.69 ± 7.50 ***^+++a3^	26.52 ± 7.19 ***^+++a3^	27.55 ± 5.59 ***^+++a3^

Statistically significant differences within groups (compared to initial level)—* *p* < 0.05; ** *p* < 0.01; *** *p* < 0.001; compared to after diet period—^+++^ *p* < 0.001; statistically significant differences between groups (compared at the same points of dynamics)—^a1^ *p* < 0.05; ^a3^ *p* < 0.001.

**Table 3 vetsci-10-00431-t003:** Serum lipid profile in control and experimental group (*n* = 7). Results are presented as mean ± SD.

Parameter	Group	Initial Level	After Diet	Day 1	Day 3	Day 5	Day 10
Total cholesterol (mmol/L)	Control	1.50 ± 0.31	1.49 ± 0.14	1.44 ± 0.22	1.47 ± 0.06	1.48 ± 0.16	1.48 ± 0.09
	Experimental	1.49 ± 0.30	2.47 ± 0.17 ^a3^	2.33 ± 0.41 ^a3^	2.34 ± 0.42	2.99 ± 0.97 *^a1^	3.07 ± 1.85 *
Triglycerides (mmol/L)	Control	0.74 ± 0.17	0.75 ± 0.26	0.71 ± 0.26	0.73 ± 0.21	0.73 ± 0.38	0.70 ± 0.33
	Experimental	0.74 ± 0.22	1.88 ± 0.33 ^a3^	3.68 ± 2.56	4.90 ± 2.38	8.32 ± 4.57 ***^++va3^	5.95 ± 3.99 *^a1^
LDL (mmol/L)	Control	0.43 ± 0.07	0.42 ± 0.06	0.41 ± 0.08	0.44 ± 0.06	0.46 ± 0.13	0.48 ± 0.06
	Experimental	0.46 ± 0.12	0.83 ± 0.11 ^a3^	0.79 ± 0.18 ^a3^	0.73 ± 0.21 ***^++vvv^	0.89 ± 0.45 ^•••a3^	0.76 ± 0.67 ^•^
VLDL (mmol/L)	Control	0.34 ± 0.08	0.34 ± 0.12	0.32 ± 0.12	0.33 ± 0.10	0.34 ± 0.17	0.32 ± 0.15
	Experimental	0.34 ± 0.10	0.86 ± 0.15 ^a3^	1.68 ± 1.17	2.24 ± 1.09	3.81 ± 2.09 ^***++va3^	2.72 ± 1.82 *^a1^
HDL (mmol/L)	Control	0.95 ± 0.23	0.94 ± 0.23	0.93 ± 0.12	0.91 ± 0.12	0.96 ± 0.25	0.93 ± 0.08
	Experimental	0.96 ± 0.25	1.24 ± 0.11 ^a1^	1.12 ± 0.24	1.02 ± 0.22	0.90 ± 0.27	0.91 ± 0.11

Statistically significant differences within groups (compared to initial level)—* *p* < 0.05; *** *p* < 0.001; compared to after diet period—^++^ *p* < 0.01; compared to day 1—^V^ *p* < 0.05; ^VVV^ *p* < 0.001; compared to day 3—^•^ *p* < 0.05; ^•••^ *p* < 0.001; statistically significant differences between groups—^a1^ *p* < 0.05; ^a3^ *p* < 0.001.

**Table 4 vetsci-10-00431-t004:** Advanced oxidation protein products (AOPP) and uric acid in serum of control and experimental group (*n* = 7). Results are presented as mean ± SD.

Parameter	Group	Initial Level	After Diet	Day 1	Day 3	Day 5	Day 10
AOPP (μmol/L)	Control	51.86 ± 7.20	63.86 ± 12.95	65.14 ± 11.94	64.86 ± 10.94	65.28 ± 9.78	67.43 ± 14.32
	Experimental	42.57 ± 7.72	66.86 ± 24.20	74.57 ± 24.18	116.57 ± 35.33 ***^+a3^	181.00 ± 44.38 ***^+++VVVa3^	113.71 ± 24.39 ***^+a3^
Uric acid (μmol/L)	Control	87.86 ± 22.12	85.43 ± 32.78	86.43 ± 18.62	87.43 ± 21.39	87.14 ± 16.60	87.00 ± 15.62
	Experimental	85.28 ± 17.88	115.00 ± 26.6	138.29 ± 21.75 *^a1^	144.57 ± 43.30 *^a1^	136.43 ± 38.06 *^a1^	129.14 ± 28.99

Statistically significant differences within groups (compared to initial level)—* *p* < 0.05; *** *p* < 0.001; compared to after diet period— ^+^ *p* < 0.05; ^+++^ *p* < 0.001; compared to day 1—^VVV^
*p* < 0.001; statistically significant differences between groups (compared at the same points of dynamics)—^a1^ *p* < 0.05; ^a3^ *p* < 0.001.

## Data Availability

The data presented in this study are available on request from the corresponding authors.

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
