# Peer review of "Metabolic Markers Associated with Progression of Type 2 Diabetes Induced by High-Fat Diet and Single Low Dose Streptozotocin in Rats"

_vetsci, 2023, doi:10.3390/vetsci10070431_

Round 1
Reviewer 1 Report (Previous Reviewer 1)
accepted
Author Response
Thank you very much for accepting our work!
Reviewer 2 Report (Previous Reviewer 2)
The manuscript entitled ”Rat model of type 2 diabetes induced by high-fat diet and low dose streptozotocin – evaluation criteria” has improved during the review process, and the aim of the study and the methods are more clearly described now. However, I still have one minor comment about the graphs that has not been adjusted yet (see comment below). All my other comments have been adequately addressed.
Minor comment
· Figure 1., Figure 2. and Figure 3: For all three of these figures the title of the Y-axis is missing. Add the parameter measured and the unit within paranthesis on the left side of the Y-axis in all three graphs.
Author Response
Point 1:
I still have one minor comment about the graphs that has not been adjusted yet (see comment below). All my other comments have been adequately addressed.
Minor comment
For all three of these figures the title of the Y-axis is missing. Add the parameter measured and the unit within paranthesis on the left side of the Y
Response 1:
The title of the Y–axis for figure 1 is added (HOMA-IR)
The title of the Y–axis for figure 2 is added (HOMA-β %)
The title of the Y–axis for figure 3 is added (Adiposity index)
Reviewer 3 Report (New Reviewer)
In this study the authors investigated various anthropometric and biochemical parameters in the first 10 days after inducing diabetes mellitus in Whistar rats. As obviously the rats were still growing, both groups, the control and experimental group, gained weight during diet period. Perhaps unexpected, parameters such as BMI, body weight and abdominal circumference were comparable between the study groups and only the adiposity index examined in a small number of rats increased significantly in the high fat diet group. As I understand these rats were neither overweight, nor obese, but had increased fat deposits? This was associated with elevated glucose concentrations compared to the controls, likely because of insulin resistance as HOMA-IR increased and HOMA-ß decreased. After injecting STZ, diabetes developed within a very short time (overt diabetes?, within one day?). Thus, this might be considered a problematic model to study metabolic markers for “progression” of type 2 diabetes.
· As this is a rat-study, the introduction should be more species specific. What do we know about biochemical parameters to identify diabetes or pre-diabetes in rats. What is the current gold standard? How are they defined? Is this the first study of this kind? Is any of the parameters new? What about glycosylated proteins or ketones?
· Did you intend to find new parameters for human diabetics (the rat as a model?), or was the idea to better classify glycemic control in rats?
· Is glucose above 12 mmol/l a universally accepted threshold for the diagnosis of diabetes mellitus in rats?
· Aside from weight loss, did the rats show clinical signs of diabetes such as polyuria/polydipsia? Did they develop overt diabetes?
· How do you define a pre-diabetic state? Do you mean the time before STZ injection? Or day one after STZ injection? Are the rats in this study diabetic or pre-diabetic? What is the renal threshold in rats? I have trouble to categorize the DR group.
· How do you define obesity?
· Were the assays used to measure biochemical parameters validated for the use in rats? CVs etc. Please add validation-data and reference ranges.
· The use of DR and experimental group and CR and control group makes reading difficult.
Introduction: Some sentences are very long (sentence 81-85) and many are difficult to understand.
Line 52: Can a disorder be activated?
Line 58: “and” lipid metabolites
Line 61: “other” anthropometric parameters, as BMI is such a parameter
Line 63: can you “confirm” glycated hemoglobin? Better: Confirm diabetes in an individual with elevated HBA1c?
Line 65 their, not its determination
Line 66-69: Sentence needs rewording.
Line 89: remove too
Line 100: evaluating or diagnosing? Sentence needs citation.
Line 102 What are the parameters currently used to diagnose T2DM in rats?
Materials and Methods
Line 115: Age?
Line 138-139: Does a glucose above 12 mmol/l confirm diabetes? Confirmed during the study, at any time? On day 1?
Lines 139 to 154: Did you use species validated assays? Within and between day precision, analytic sensitivity?
Line 184: Data normally distributed?
Line 328: Neutrophils or lymphocytes?
Line 340 STZ not streptozotocin
Lines 342: parenchyma
Lines 354 to 373: It sounds as if you were able to induce diabetes very fast by feeding a high fat diet alone. Don’t you think that STZ introduces many additional problems that have to be addressed.
Lines 374 to 378: Why did you use growing animals? This paragraph should focus on the main conclusion, that the BMI or abdominal circumference are insensitive markers for obesity in rats.
Line 428: clearance of blood serum: from triglycerides?
Line 453: Why dose uric acid increase in diabetic rats?
The sentences are long and difficult to understand.
Author Response
Point 1:
In this study the authors investigated various anthropometric and biochemical parameters in the first 10 days after inducing diabetes mellitus in Wistar rats.
Response 1:
The experimental model allows us to focus on two key moments in the development of type 2 diabetes in male Wistar rats.
The first concerns the end of the 4-week high-fat diet (before STZ administration) when AI is increased (Figure 3), glucose and insulin concentrations increase (Table 2), dyslipidemia (Table 3) and insulin resistance are present (Figure 1). These data characterize a pre-diabetic state associated with obesity. Unfortunately, no standardized definition of obesity exists in rats. However, obesity is an important predisposing factor for the development of type 2 diabetes. It may not automatically or immediately lead to the development of type 2 diabetes.
The transition to overt diabetes, however, requires a disruption of the functional ß-cells, which is achieved by the single administration of a low dose of STZ. The focus of monitoring (until the 10th day) aims to emphasize early changes in carbohydrate, lipid and protein metabolism, so as to differentiate metabolic markers.
Point 2:
As obviously the rats were still growing, both groups, the control and experimental group, gained weight during diet period.
Response 2:
The rats used in the study were 6-7 weeks old, so their growth continued throughout the study period. Reed et al., 2000, whose experimental model of type 2 diabetes is similar to ours also used 7-week-old Sprague-Dawley rats.
Our rationale for using young animals is that values of biochemical parameters in these animals show less individual variations (lower coefficient of variation) compared to older animals (Charles River, 1993). In older animals, the high degree of heterogeneity is due to age-related changes, which may make the study less informative. Han et al., 2020 also reported significant inter-individual variation in blood glucose at 22–26 weeks of age.
Point 3:
Perhaps unexpected, parameters such as BMI, body weight and abdominal circumference were comparable between the study groups and only the adiposity index examined in a small number of rats increased significantly in the high fat diet group. As I understand these rats were neither overweight, nor obese, but had increased fat deposits? This was associated with elevated glucose concentrations compared to the controls, likely because of insulin resistance as HOMA-IR increased and HOMA-ß decreased.
Response 3:
Body weight alone is not an indicator of obesity in rats therefore BMI is also not a reliable measure of obesity in this species.
Huang et al. 2005 (Br J Pharmacol 145: 767–774) found that despite lower body weight in diabetic rats, visceral fat accumulation persisted.
In our study, the body weight of diabetic rats was similar to that of controls, but their AI increased, which was due to an increase in abdominal adipose tissue. Already at the end of the high-fat diet, AI shows an increase which leads to the conclusion that this indicator can be an objective marker of obesity.
Therefore, in the discussion we note (line 391 – 392) that the calculation of AI and the evaluation of visceral fat are an important and reliable indicator of obesity in this biological species, in contrast to the recently used BMI.
Point 4:
After injecting STZ, diabetes developed within a very short time (overt diabetes?, within one day?). Thus, this might be considered a problematic model to study metabolic markers for “progression” of type 2 diabetes.
Response 4:
In our opinion, the Wistar rat model of type 2 diabetes, based on the development of dietary induced insulin resistance (after 4 weeks on high-fat diet) and single low-dose STZ administration (HFD/STZ model) is not problematic, but advantageous because of the following reasons:
- the model allows to reveal the natural progression of diabetic pathology from pre-diabetes (insulin resistance associated with obesity) to type 2 diabetes in a short period of time;
- the model allows to identify early metabolic markers related to carbohydrate, lipid and protein metabolism in diabetic rats;
- the model allows to study the role of oxidative stress in the pathology of diabetes by analyzing the changes in some markers of oxidative stress - AOPP, uric acid. Oxide radicals play an important role in the pathogenesis of diabetes, as the diabetic hyperglycemia stimulates their increase.
The dose of STZ is a crucial factor. It attacks pancreatic ß-cells by entering them via GLUT 2. It induces a steady but not absolute loss of ß-cell mass. On one hand, we use a single low dose of STZ and on the other hand, we use rats aged 6-7 weeks. These young rodents have the ability to increase ß-cell mass. This approach allows the rapid induction of ß-cell death induced by the single dose of STZ to be blunted to some extent.
Point 5:
As this is a rat-study, the introduction should be more species specific.
Response 5:
The following text is added in the Introduction: “Despite the existence of a large number of studies related to the modeling of T2DM in rats by means of HFD and STZ, there are no sufficiently clear criteria for evaluating the different stages of the progression of diabetes in this biological species. Skovsø [30] discusses extensively the specific variations in dietary regimen, age, dose of STZ when using HFD/STZ rat models. This researcher also discusses metabolic parameters (insulin, glucose, total cholesterol, triglycerides, LDL, HDL, HOMA-IR, HbA1c) used to identify the stage of T2DM in rats”.
Point 6:
What do we know about biochemical parameters to identify diabetes or pre-diabetes in rats.
Response 6:
Glucose and insulin concentrations in rats are the most commonly reported time-to-diabetes criteria (Peterson et al., 2015). Pre-diabetes often includes a state of obesity characterized by insulin resistance and dyslipidemia (Tabak et al., 2009). Essentially, however, the transition from a pre-diabetic state to overt diabetes requires the loss of a significant proportion of functional β-cell mass (Prentki and Nolan, 2006).
Point 7:
What is the current gold standard? How are they defined?
Response 7:
There is no established gold standard for rats. Literature data show that BMI and metabolic indicators - insulin, glucose, total cholesterol, triglycerides, LDL are most often measured.
Point 8:
Is this the first study of this kind?
Response 8:
In this study, an experimental model of type 2 diabetes was used in order to make an integral assessment of the development of the disease by tracking the changes of anthropometric parameters (body weight, body length, BMI, abdominal circumference), some indices (HOMA-IR, HOMA-β, AI), metabolic parameters of carbohydrate and lipid metabolism, purine metabolites, protein products of advanced oxidation (AOPP) and histologic changes of liver and pancreas. The aim is to select the most reliable evaluation criteria suitable for this biological species.
Point 9:
Is any of the parameters new?
Response 9:
The use of the parameters uric acid and AOPP to study the early stages of the development of type 2 diabetes in rats is an important novel point, as it allows to evaluate the role of oxidative stress in the progression of the disease. These indicators, which are one of the markers of oxidative stress, are not new, but they are easily accessible and their use gives more complete information about their involvement in the development of diabetes in rats.
Point 10:
What about glycosylated proteins or ketones?
Response 10:
Non-enzymatic glycosylation, a process in which glucose is chemically attached to an amino group of proteins without the aid of enzymes, characterizes the complications of diabetes mellitus. Glycosylation is an important pathogenetic factor in the complications of diabetes. Because we focus on the early stage, the products of this glycosylation are not considered.
Point 11:
- Did you intend to find new parameters for human diabetics (the rat as a model?), or was the idea to better classify glycemic control in rats?
Response 11:
The idea of this study is to provide a suitable algorithm for the development of type 2 diabetes in Wistar rats, which are often used in research.
Point 12:
- Is glucose above 12 mmol/l a universally accepted threshold for the diagnosis of diabetes mellitus in rats?
Response 12:
Yes, glucose levels above 12 mmol/l characterize hyperglycemia in rats [38].
Point 13:
Aside from weight loss, did the rats show clinical signs of diabetes such as polyuria/polydipsia? Did they develop overt diabetes?
Response 13:
These clinical signs are manifestations of metabolic disturbances and are observed in rats, but we did not measure them.
Point 14:
How do you define a pre-diabetic state? Do you mean the time before STZ injection? Or day one after STZ injection? Are the rats in this study diabetic or pre-diabetic?
Response 14:
The data obtained at the end of the high-fat diet are commented on as pre-diabetes, and overt diabetes occurs after the administration of STZ.
Point 15:
What is the renal threshold in rats?
Response 15:
Hutter, 2019 (Doctoral thesis, University of Zurich; Zurich Open Repository and Archive. https://doi.org/10.5167/uzh-172302) comments on the role of blood glucose concentrations in polydipsia and polyuria.
Point 16:
I have trouble to categorize the DR group.
Response 16:
DR is the experimental group, which is now corrected in the text.
Point 17:
- How do you define obesity?
Response 17:
There is no exact definition of obesity in rats. We used AI, HOMA-IR to prove it.
Point 18:
- Were the assays used to measure biochemical parameters validated for the use in rats? CVs etc. Please add validation-data and reference ranges.
Response 18:
All assays in the current study were performed in a certified lab in compliance with good laboratory practices and internal quality control procedures: 1) intra-laboratory reproducibility: at least six measurements of control serum samples (normal and pathological) performed on different days and with different batches of the reagents; 2) utilization of calibration curves with coefficient of correlation of at least r=0.98; 3) performance of tests in line with manufacturer's instructions; 4) For ELISA tests - multiple testing of positive and negative control samples throughout the day; performing all samples in duplicate.
Point 19:
- The use of DR and experimental group and CR and control group makes reading difficult.
Response 19:
It is now corrected in the text.
Point 20:
Introduction: Some sentences are very long (sentence 81-85) and many are difficult to understand.
Corrected
Line 52: Can a disorder be activated?
Corrected
Line 58: “and” lipid metabolites
Corrected
Line 61: “other” anthropometric parameters, as BMI is such a parameter
Corrected
Line 63: can you “confirm” glycated hemoglobin? Better: Confirm diabetes in an individual with elevated HBA1c?
Corrected
Line 65: their, not its determination
Corrected
Line 66-69: Sentence needs rewording.
Corrected
Line 89: remove too
Corrected
Line 100: evaluating or diagnosing? Sentence needs citation.
Corrected
Point 21
Materials and Methods
Line 115: Age?
Information is added.
Line 138-139: Does a glucose above 12 mmol/l confirm diabetes? Confirmed during the study, at any time? On day 1?
Information is added.
Lines 139 to 154: Did you use species validated assays? Within and between day precision, analytic sensitivity?
Insulin ELISA kit used in the study was rat-specific. Its Inter-Assay CV was <8% and intra-Assay CV: <10%; For colorimetric assays, there is no need from utilizing species-specific analytical kits
Line 184: Data normally distributed?
Information is added.
Line 328: Neutrophils or lymphocytes?
Leukocytes infiltration
Line 340: STZ not streptozotocin
Corrected
Lines 342: parenchyma
Corrected
Lines 354 to 373: It sounds as if you were able to induce diabetes very fast by feeding a high fat diet alone. Don’t you think that STZ introduces many additional problems that have to be addressed.
Pre-diabetes often includes a state of obesity characterized by insulin resistance and dyslipidemia (Tabak et al., 2009). Essentially, however, the transition from a pre-diabetic state to overt diabetes requires the loss of a significant proportion of functional β-cell mass (Prentki and Nolan, 2006).
Lines 374 to 378: Why did you use growing animals?
In general, clinical chemistry values of younger rats (less than 6 months of age) show less interindividual variability (lower coefficient of variation (CV)%) than older animals (Charles River, Inc., 1993). In older animals, the high degree of heterogeneity is due to age-related changes and spontaneous disease may obscure the effects of the compound, making clinicopathological examination less informative.
Line 428: clearance of blood serum: from triglycerides?
Corrected
Line 453: Why dose uric acid increase in diabetic rats?
Explained
Round 2
Reviewer 3 Report (New Reviewer)
Thank you very much for addressing most of my comments.
1.) Diabetes was diagnosed by measuring high glucose concentrations. How many glucose measurements had to be above 12 mmol/l in each rat to diagnose diabetes. Was glucose measured after an overnight fast? Is hyperglycemia equal to diabetes? What about stress-hyperglycemia?
2.) One of the markers tested was uric acid. Why do the authors think that uric acid increased in their rats? Increased production or reduced clearance? Is it possible that STZ caused kidney damage and associated reduced clearance? Uric acid increases in nephropathy in rats! If uric acid increases due to diabetes itself, should it no longer be used as a marker for diabetic nephropathy? Is this increase seen in other diabetic models, in other studies?
3.) Leukocyte infiltrates- what was the dominant cell type? neutrophil, lymphocytes, macrophages....?
4.) To determine Adiposity Index the animals had to be sacrificed. Cole et al. tested for muscle mass using DEXA! Do I understand it correct that currently no validated sensitive method exists that allows the diagnosis of obesity in live mice?
Line 419: statistically significant
Line 151-153: Sentence does not make sense.
Has improved, but I am no native speaker!
Author Response
Response to reviewer 3 (Round 2)
Point 1:
1.) Diabetes was diagnosed by measuring high glucose concentrations. How many glucose measurements had to be above 12 mmol/l in each rat to diagnose diabetes. Was glucose measured after an overnight fast? Is hyperglycemia equal to diabetes? What about stress-hyperglycemia?
Response 1:
- How many glucose measurements had to be above 12 mmol/l in each rat to diagnose diabetes?
One measurement. All STZ treated rats in our study had values much higher than 12 mmol/l. Values measured on the following days (1, 3, 5 & 10) also confirm diabetes.
According to Qinna and Baadwan, after STZ administration “rats with a basal blood glucose level above 200 mg/dL were considered diabetic unless otherwise stated” [38].
- Was glucose measured after an overnight fast?
Yes, glucose was measured after an overnight fast.
- Is hyperglycemia equal to diabetes? What about stress-hyperglycemia?
We believe that hyperglycemia is diabetes-related rather than stress-induced because of the following:
- Animals from the control group had no statistically significant deviations in glucose levels.
- To avoid stress all rats had a 2-week adaptation period, during which they had multiple contacts with the personnel and became adapted to laboratory environment.
Point 2:
2.) One of the markers tested was uric acid. Why do the authors think that uric acid increased in their rats? Increased production or reduced clearance? Is it possible that STZ caused kidney damage and associated reduced clearance? Uric acid increases in nephropathy in rats! If uric acid increases due to diabetes itself, should it no longer be used as a marker for diabetic nephropathy? Is this increase seen in other diabetic models, in other studies?
Response 2:
It is difficult to give an unequivocal answer to this question. According to Wu et al., (Hindawi Evidence-Based Complementary and Alternative Medicine, 2018 https://doi.org/10.1155/2018/6821387) the pathogenesis of hyperuricemia is complicated. In the “Results” section of the article the author states: STZ increases uric acid excretion and reduces uric acid production in hyperuricemic rats. On the other hand Rahman et al. 2023 (https://doi.org/10.1038/s41598-023-29445-8) found an increase in uric acid in rats with induced diabetes, as we also found.
Point 3:
3.) Leukocyte infiltrates- what was the dominant cell type? neutrophil, lymphocytes, macrophages....?
Response 3:
Neutrophils were the dominant cell type. The information is now added in the manuscript.
Point 4:
4.) To determine Adiposity Index the animals had to be sacrificed. Cole et al. tested for muscle mass using DEXA! Do I understand it correct that currently no validated sensitive method exists that allows the diagnosis of obesity in live mice?
Response 4:
Only Cole et al. have used DEXA in mice. Mariana de Moura e Dias et al. (Diabetol Metab Syndr., 2021, 13, 32) evaluated mice and rat models published in the last 6 years. They have stated that adiposity index is the main indicator, which corresponds to visceral fat and is now considered to be a criterion for obesity in experimental rodents.
Point 5:
Line 419: statistically significant
Response 5:
Corrected
Point 6:
Line 151-153: Sentence does not make sense.
Response 6:
Corrected
This manuscript is a resubmission of an earlier submission. The following is a list of the peer review reports and author responses from that submission.
Round 1
Reviewer 1 Report
English language and style are fine/minor spell check required
Author Response
Response to reviewer 1 comments
Point 1: English language and style are fine/minor spell check required.
Response 1: Thank you for your review. Spelling mistakes have been corrected.
Reviewer 2 Report
The manuscript entitled ”Rat model of type 2 diabetes induced by high-fat diet and low dose streptozotocin – evaluation criteria” describes that the combined treatment with high-fat diet and low dose streptozotocin mimicks the insulin resistance and beta-cell dysfunction seen in patients with type 2 diabetes, indicating that this is usuful animal model in type 2 diabetes research. Similar models for type 2 diabetes, combining high-fat diet and streptozotocin have previously been published, so the idea is not completely new. However, the authors of the present paper have made thorough analysis of metabolic markers for both glucose and lipid metabolism to evaluate the model.
Comments:
Page 3, Line 134: Here it says “Diabetes was confirmed by blood glucose ≥ 12 mmol/L and rats were used in the present study”, but it’s not clear how many of the 35 rats had confirmed diabetes according to these criteria. I suggest yo add “Diabetes was confirmed in all STZ-injected rats by blood glucose…” or add the number of rats that had confirmed diabetes in the STZ-injected group.
Page 3, Line 133: Here it says that blood samples were taken from the tail vein to measure glucose levels by Glucomer one day after STZ injection but further down on the same page (Line 137) it says that blood samples were collected by retroorbital bleeding to at all different stages of the study including 1 day after STZ injection to measure blood glucose by another method (enzymatic colorimetric test).
A) Why did you measure blood glucose with two different methods?
B) And data from which glucose method is presented in Table 2., Figure 1. and Figure 2.?
C) Furthermore, it’s not stated if the rats were fasted and for how long they were fasted before the glucose and insulin sampling, which they should have been if you use the data for HOMA-IR and HOMA-β calculations.
D) On Line 137 it doesn’t say how many rats were sampled by retro-orbital bleeding at each time point, and if it was done in association with termination or if reapeted blood samples were taken from the same rats.
Page 3, Line 146: Here it only says that “VLDL – calculated from Friedewald’s formula;” but no formula is written out and no reference cited showing that this formula is relevant in rats.
Page 5, Table 2. and Page 7, Table 3.: In Table 2&3. it’s a bit confusing when the numbers for the experimental group ends up at different rows depending on if there are statsitical signs after the numbers or not. This could maybe be fixed by changing the layout?
Page 6, Figure 1. and Figure 2.: For both these figures the title of the Y-axis is missing and the explanation of the statistical signs is missing in the legend.
Page 7, Figure 3.: For Figure 3. the title of the Y-axis, explanation of which line is representing which group, error bars and explanation of statistical signs are missing.
Page 9, Line 318-332: In this paragraph the design of the present study is discussed, but I’m missing a discussion about how this model differs from previous rat models where high fat diet and STZ-injections have been combined (for example ref 37). What is the novelty/advantages of your specific study design?
Page 10, Lines 346-355: In this paragraph the authours discuss that calculation of adiposity index (AI) is the most objective criterion for the actual degree of obesity. However, nothing is mentioned about measuring fat % by DEXA or MR, which are two frequently used reliable methods to measure body composition in rodents. I understand that these methods were not used in this study, but should be included in the discussion of measurements for obesity.
Page 11, Line 390: Here the association between changes in the lipid profile and increase in serum concentration of free fatty acids is pointed out. It would have been interesting to see a measurement of the free fatty acids in the serum of the rats in the present study as well. Is there any possibility to include these measurements in the manuscript?
Page 11, Line 424: The Declaration of Helsinki is not relevant here since this is not a study including human subjects. The ARRIVE guidelines or the EU regulations on animal research are more relevant in this case.
· Page 11, Line 427: I guess the informed consent statement has been included by mistake, or did you ask all the rats in the study to sign an informed consent? :-)
Author Response
Response to reviewer 2 comments
Point 1: Page 3, Line 134: Here it says “Diabetes was confirmed by blood glucose ≥ 12 mmol/L and rats were used in the present study”, but it’s not clear how many of the 35 rats had confirmed diabetes according to these criteria. I suggest yo add “Diabetes was confirmed in all STZ-injected rats by blood glucose…” or add the number of rats that had confirmed diabetes in the STZ-injected group.
Response 1: Your suggestion was added to the text of the revised manuscript: “Diabetes was confirmed in all STZ-injected rats by blood glucose…”
Point 2: Page 3, Line 133: Here it says that blood samples were taken from the tail vein to measure glucose levels by Glucomer one day after STZ injection but further down on the same page (Line 137) it says that blood samples were collected by retroorbital bleeding to at all different stages of the study including 1 day after STZ injection to measure blood glucose by another method (enzymatic colorimetric test).
- A)Why did you measure blood glucose with two different methods?
- B)And data from which glucose method is presented in Table 2., Figure 1. and Figure 2.?
- C)Furthermore, it’s not stated if the rats were fasted and for how long they were fasted before the glucose and insulin sampling, which they should have been if you use the data for HOMA-IR and HOMA-β calculations.
- D)On Line 137it doesn’t say how many rats were sampled by retro-orbital bleeding at each time point, and if it was done in association with termination or if reapeted blood samples were taken from the same rats.
Response 2:
- A) We used glucometer and a blood sample from the tail vain only to confirm diabetes in STZ-injected group. For all other measurements we used the enzymatic colorimetric test after retroorbital bleeding.
- B) Values presented in table 2 and figures 1&2 were obtained from the enzymatic colorimetric test.
- C) Rats were fasted overnight before the glucose and insulin sampling.
- D) At each time point a single blood samplewas collected from 7 rats.
Point 3: Page 3, Line 146: Here it only says that “VLDL – calculated from Friedewald’s formula;” but no formula is written out and no reference cited showing that this formula is relevant in rats.
Response 3: The Friedewald’s formula is added along with a reference.
Point 4: Page 5, Table 2. and Page 7, Table 3.: In Table 2&3. it’s a bit confusing when the numbers for the experimental group ends up at different rows depending on if there are statsitical signs after the numbers or not. This could maybe be fixed by changing the layout?
Response 4: The layout of the tables is now changed in the revised version of the manuscript.
Point 5: Page 6, Figure 1. and Figure 2.: For both these figures the title of the Y-axis is missing and the explanation of the statistical signs is missing in the legend.
Response 5: The missing components are now added in the revised version of the manuscript.
Point 6: Page 7, Figure 3.: For Figure 3. the title of the Y-axis, explanation of which line is representing which group, error bars and explanation of statistical signs are missing.
Response 6: The missing components are now added in the revised version of the manuscript.
Point 7: Page 9, Line 318-332: In this paragraph the design of the present study is discussed, but I’m missing a discussion about how this model differs from previous rat models where high fat diet and STZ-injections have been combined (for example ref 37). What is the novelty/advantages of your specific study design?
Response 7: The advantage of our experimental model is the short duration of the high-fat diet. Some studies [like the above mentioned ref 37] use even a shorter period of high-fat diet, but without changes in the blood glucose concentrations. Our study demonstrates that 4 weeks on a high-fat diet (calories from fat > 40%) can disrupt the glucose metabolism. Moreover, high-fat diets with fat higher than 50% have influence on the severity of metabolic disorders and β-cellular function [35,43].
Point 8: Page 10, Lines 346-355: In this paragraph the authours discuss that calculation of adiposity index (AI) is the most objective criterion for the actual degree of obesity. However, nothing is mentioned about measuring fat % by DEXA or MR, which are two frequently used reliable methods to measure body composition in rodents. I understand that these methods were not used in this study, but should be included in the discussion of measurements for obesity.
Response 8: New comment is now included in the discussion of the revised version of the manuscript.
“Dual energy x-ray absorptiometry (DEXA) is currently widely used in humans, as one of the most precise noninvasive methods for analysis of body composition. Cole et al., 2020 [49] has demonstrated the use of DEXA in mice.”
Point 9: Page 11, Line 390: Here the association between changes in the lipid profile and increase in serum concentration of free fatty acids is pointed out. It would have been interesting to see a measurement of the free fatty acids in the serum of the rats in the present study as well. Is there any possibility to include these measurements in the manuscript?
Response 9: Unfortunately concentration of free fatty acids was not measured!
Point 10: Page 11, Line 424: The Declaration of Helsinki is not relevant here since this is not a study including human subjects. The ARRIVE guidelines or the EU regulations on animal research are more relevant in this case.
Response 10: Your comment is absolutely correct. Relevant information is added in the revised version.
“The study was conducted in accordance with Directive 63/2010/EU, and approved by the Ethics Committee of the Bulgarian Food Safety Agency (Permit No. 281; opinion of the Ethics Committee No. 197 of 10.09.2020)”.
Point 11: Page 11, Line 427: I guess the informed consent statement has been included by mistake, or did you ask all the rats in the study to sign an informed consent? :-)
Response 11: Of course, the informed consent statement was included by mistake! J
Reviewer 3 Report
I’m afraid I do not understand why this study was conducted. The authors write that ‘The use of experimental animal models is an important tool in understanding the pathogenesis of T2DM.’ But we already understand how human type 2 diabetes develops and how to prevent it: with diet, physical activity and sustained weight loss. Indeed, this paper acknowledges lifestyle factors: ‘It has been proven that unhealthy eating habits, inactivity, obesity, high levels of non-esterified fatty acids, glycerol, hormonal imbalance, hypoxia, oxidative stress, as well as genetic factors are involved in the pathogenesis of T2DM in humans.’ And if there are any outstanding questions (‘A number of cellular pathogenetic mechanisms still re- 90 main insufficiently elucidated’) these would be better studied in humans, or in human biology-based models. Type 2 diabetes is increasingly studied using organ-on-a-chip technology, which has the advantage of being directly relevant to humans and avoiding the risk of poor extrapolation from animal models to humans. But even more importantly, we already know how to prevent T2DM or induce remission in humans. Indeed, Unwin et al (2023) recently found that patients with T2DM at a general practice in Southport were able to achieve remission of their disease simply by losing weight and following a low carbohydrate diet. The patients were healthier and the surgery spent only £4.94 per patient each year on diabetes drugs compared with the £11.30 per patient spent by neighbouring practices. Therefore I fail to see the need for an animal model of T2DM and frankly, I’m surprised this project gained ethical approval.
Unwin D, Delon C, Unwin J, Tobin S, Taylor R. What predicts drug-free type 2 diabetes remission? Insights from an 8-year general practice service evaluation of a lower carbohydrate diet with weight loss. BMJ Nutr Prev Heal. Published online January 2, 2023:e000544. doi:10.1136/bmjnph-2022-000544
Author Response
Response to reviewer 3 comments
Point 1: I’m afraid I do not understand why this study was conducted. The authors write that ‘The use of experimental animal models is an important tool in understanding the pathogenesis of T2DM.’ But we already understand how human type 2 diabetes develops and how to prevent it: with diet, physical activity and sustained weight loss. Indeed, this paper acknowledges lifestyle factors: ‘It has been proven that unhealthy eating habits, inactivity, obesity, high levels of non-esterified fatty acids, glycerol, hormonal imbalance, hypoxia, oxidative stress, as well as genetic factors are involved in the pathogenesis of T2DM in humans.’ And if there are any outstanding questions (‘A number of cellular pathogenetic mechanisms still re- 90 main insufficiently elucidated’) these would be better studied in humans, or in human biology-based models. Type 2 diabetes is increasingly studied using organ-on-a-chip technology, which has the advantage of being directly relevant to humans and avoiding the risk of poor extrapolation from animal models to humans. But even more importantly, we already know how to prevent T2DM or induce remission in humans. Indeed, Unwin et al (2023) recently found that patients with T2DM at a general practice in Southport were able to achieve remission of their disease simply by losing weight and following a low carbohydrate diet. The patients were healthier and the surgery spent only £4.94 per patient each year on diabetes drugs compared with the £11.30 per patient spent by neighbouring practices. Therefore I fail to see the need for an animal model of T2DM and frankly, I’m surprised this project gained ethical approval.
Response 1: Experimental animal models are still an important tool in diabetology. Their use continues because they provide valuable information about the complex pathogenetic mechanisms in type 2 diabetes. Thanks to an experimental model with rats subjected to a 2-week high-fat diet and streptozotocin, Young J. [2021] found that not only pancreatic β-cells, but hypothalamic neurons also exert a controlling influence on blood sugar: "Astrocytes function as glucose sensors in the brain"; „Leptin-sensitive, glucose regulating neurons become resistant to leptin during aging or during exposure to a high-fat diet. These leptin resistant neurons fail to restrain food intake, obesity, and blood glucose. The reasons for this lowered responsiveness to leptin are uncertain and are part of the puzzle of the causes of type II DM”; “However, since the same type of astrocyte has been identified in the same region of the human hypothalamus, it is plausible that these findings can be extended to an understanding of human brain function”.
In our opinion animal experimental models should be continuously improved. The advantage of our experimental model is the short duration of the high-fat diet. Some studies [ref 37] use even a shorter period of high-fat diet, but without changes in the blood glucose concentrations. Our study demonstrates that 4 weeks on a high-fat diet (calories from fat > 40%) can disrupt the glucose metabolism. Moreover, high-fat diets with fat higher than 50% have influence on the severity of metabolic disorders and β-cellular function [35,43].
Round 2
Reviewer 3 Report
Thank you for sending me the revised manuscript and the response to my review.
Unfortunately I don't consider the response to be adequate and there is no acknowledgement in the revised manuscript of the problem of unreliable extrapolation from animals to humans. The authors defend their approach by stating that animal models provide valuable information about the complex pathogenetic mechanisms in type 2 diabetes. That may well be true, but unfortunately that information is only relevant to rats, not humans. They refer to research by Young (2021), but this only proves my point, that extrapolation is unreliable, i.e.: 'However, since the same type of astrocyte has been identified in the same region of the human hypothalamus, it is plausible that these findings can be extended to an understanding of human brain function'. The word 'plausible' says it all. In fact, decades of research on rats has provided ample evidence that it is unlikely that findings in rats can be reliably extended to humans. This paper by Robert Perlman, explains why extrapolation from rodents to humans is so unreliable: Mouse models of human disease. An evolutionary perspective. 2016. https://doi.org/10.1093/emph/eow014
Furthermore, the authors state that it is their 'opinion' that experimental animal models should be continuously improved. But again, there is no evidence that improvement of animal models leads to better translation to humans.
The authors state at the end of their manuscript:
Institutional Review Board Statement: The study was conducted in accordance with Directive 2010/63/EU, and approved by the Ethics Committee of the Bulgarian Food 449 Safety Agency (Permit No. 281; opinion of the Ethics Committee No. 197 of 10. 09. 2020).
I'm unclear in what way the study was conducted in accordance with Directive 2010/63/EU? This Directive explicitly states: ‘The use of animals for scientific or educational purposes should (…) only be considered where a non-animal alternative is unavailable.’ As I stated in my review, there are non-animal alternatives available.
Finally, the authors do not explain why it is important to understand the 'complex pathogenetic mechanisms in type 2 diabetes' if we already understand its pathogenesis and know how to avoid or achieve remission of this disease in humans.
Author Response
The “Introduction” of the manuscript was completely revised and hopefully improved. We have added the references recommended by you – Unwin et al. [15] and Perlman [20]. We have also added new relevant references providing background about etiology and pathogenesis of diabetes in animals and the use of various research experimental models including non-animal models [1,4,7,8,22]. In the conclusion, we have mentioned the main limitation of using rat models.
We agree with the conclusions of Robert Perlman in his paper “Mouse models of human disease. An evolutionary perspective” regarding the reliability of mouse models in human medicine reserach. Anyhow, Perman concludes that “For reasons mentioned above, research on mice (and other species) is essential and should be supported”.
The “Institutional Review Board Statement” was corrected.